# Development of Inactivated FAKHRAVAC^®^ Vaccine against SARS-CoV-2 Virus: Preclinical Study in Animal Models

**DOI:** 10.3390/vaccines9111271

**Published:** 2021-11-03

**Authors:** Soheil Ghasemi, Kosar Naderi Saffar, Firooz Ebrahimi, Pezhman Khatami, Arina Monazah, Ghorban-Ali Alizadeh, Hossein-Ali Ettehadi, Iman Rad, Shahrzad Nojehdehi, Mousa Kehtari, Fatemeh Kouhkan, Hesam Barjasteh, Sohrab Moradi, Mohammad-Hosein Ghorbani, Ali Khodaie, Moslem Papizadeh, Roghayeh Najafi, Ehsan Naghneh, Davood Sadeghi, Ahmad Karimi Rahjerdi

**Affiliations:** 1Milad Daro Noor Pharmaceutical (MDNP) Company, Unit 3, Mirsharifi Alley, Valiasr Street, Tehran 1986936914, Iran; S-Ghasemi@mdnp.ir (S.G.); K-Naderi@mdnp.ir (K.N.S.); P-Khatami@mndp.ir (P.K.); A-Monazah@mndp.ir (A.M.); GA-Alizadeh@mndp.ir (G.-A.A.); HA-Ettehadi@mndp.ir (H.-A.E.); H-Barjasteh@mdnp.ir (H.B.); S-Moradi@mdnp.ir (S.M.); MH-Ghorbani@mdnp.ir (M.-H.G.); A-Khodaie@mdnp.ir (A.K.); M-Papizadeh@mdnp.ir (M.P.); R-Najafi@mdnp.ir (R.N.); E-Naghneh@mdnp.ir (E.N.); 2Stem Cell Technology Research Center (STRC), Building No. 9, 2nd East Alley, Mohammad-Ali Keshavarz Blvd., Saadat Abad, Tehran 1997775555, Iran; rad@strc.ac.ir (I.R.); sh.nojehdehi@strc.ac.ir (S.N.); kehtari@strc.ac.ir (M.K.); f.kouhkan@yahoo.com (F.K.); 3Department of Biology, Faculty of Basic Sciences, Imam Hussein University, Tehran 1698715461, Iran; ebrahimi@ihu.ac.ir (F.E.); sadeghi@ihu.ac.ir (D.S.)

**Keywords:** SARS-CoV-2, vaccine candidate, FAKHRAVAC, COVID-19

## Abstract

The recent viral infection disease pandemic, caused by severe acute respiratory syndrome coronavirus 2 (SARS-CoV-2), has resulted in a global public health crisis. Iran, as one of the countries that reported over five million infected cases by September 2021, has been concerned with the urgent development of effective vaccines against SARS-CoV-2. In this paper, we report the results of a study on potency and safety of an inactivated SARS-CoV-2 vaccine candidate (FAKHRAVAC) in a preclinical study so as to confirm its potential for further clinical evaluation. Here, we developed a pilot-scale production of FAKHRAVAC, a purified inactivated SARS-CoV-2 virus vaccine candidate that induces neutralizing antibodies in Balb/c mice, guinea pigs, rabbits, and non-human primates (Rhesus macaques—RM). After obtaining ethical code of IR.IUMS.REC.1399.566, immunizations of animals were conducted by using either of three different vaccine dilutions; High (H): 10 μg/dose, Medium (M): 5 μg/dose, and Low (L): 1 μg/dose, respectively. In the process of screening for viral seeds, viral strains that resulted in the most severe clinical manifestation in patients have been isolated for vaccine development. The viral seed produced the optimal immunity against SARS-CoV-2 virus, which suggests a possible broader neutralizing ability against SARS-CoV-2 strains. The seroconversion rate at the H-, M-, and L-dose groups of all tested animals reached 100% by 28 days after immunization. These data support the eligibility of FAKHRAVAC vaccine candidate for further evaluation in a clinical trial.

## 1. Introduction

One year after the beginning of COVID-19’s global spread in December of 2019, more than 160 vaccine candidates have been introduced globally. Studies on the production of vaccine candidates is still going on in different platforms of RNA, DNA, protein-subunit, non-replicating/replicating viral vectors, and both live attenuated and inactivated vaccines [1]. Three out of ten vaccine candidates that developed to third-phase of clinical trial by end of 2020 were inactivated vaccines including the Sinovac, Bharat Biotech, and Wuhan institute of biological products [2,3,4].

Vaccine-induced disease enhancement has been a concern for vaccine developers in the past 50 years. For example, vaccination with inactivated respiratory syncytial virus (RSV) in alum resulted in enhanced respiratory disease (ERD) as a type of antibody-dependent enhancement (ADE) due to Th_1_/Th_2_ imbalance in cellular immunity upon subsequent RSV infection. During monitoring of inactivated vaccine candidate via two-dose immunization program in this study, no observable antibody-dependent enhancement (ADE) or immune-pathological exacerbation was observed in vaccinated *Rhesus macaques* (RMs). However, the development of a safe vaccine against SARS-CoV-2 (and future SARS-CoVs) without vaccine-mediated ADE is still of great importance [2,3,5].

Here, we report on the potency and safety of an inactivated SARS-CoV-2 vaccine candidate (FAKHRAVAC) in a preclinical study to confirm its potential for further clinical evaluation.

## 2. Experimental Model

All animals involved in this study were housed and fed *ad-libidum* in an association with Iran University of medical sciences (IUMS), with ethics committee reference number of IR.IUMS.REC.1399.566. All experimental procedures with mice, rats, rabbits, and non-human primates (*Rhesus macaque*; *RM*) were performed in an animal biosafety level-3 laboratory (ABSL-3).

### 2.1. Animal Models

The Balb/C mice (18–22 g), guinea pigs (250–350 g), chinchilla rabbits (1.5 kg), and *Rhesus macaques* (two- to four-year-old) were kept at 22 ± 2 °C with constant humidity and provided with a twelve hours of light/dark cycles in BSL3 animal room.

### 2.2. Isolation of Viral Strain

SARS-CoV-2 strains that were isolated from oropharynx swabs of hospitalized patients were screened to find an optimal viral seed. The obtained strains were isolated from patients who had severe clinical demonstrations such as fever, seizure, muscle cramp and reduction of arterial oxygen saturation. In a parallel strategy, the rate of infection of the patient with SARS-CoV-2, caused by recent COVID-19 outbreak, was evaluated via qRT-PCR method. The lower the Ct (cycle threshold) of the qRT-PCR method, the higher the amount of viral load in the host (the patient that the virus was isolated from at the first place). Vero cells (Cat. # 88020401), as WHO certified cell line for vaccine production, were used to replicate the five viral strains. One of the isolated strains from Vero cells that replicated the most and resulted in highest virus yields among other strains has been chosen for further development of the Milad Daro Noor Pharmaceutical (MDNP) company’s “SARS-CoV-2 inactivated vaccine” (FAKHRAVAC; formerly named as MIVAC).

### 2.3. Viral Titration

Pre-cultured Vero cells in 12-well plates (Sigma C1008) were infected with an isolated viral strain, which previously had resulted in the highest virus yield (Ct < 20 in qRT-PCR method). Then, ten-step serial dilution of the purified viral replicates stock was prepared. Vero E6 cells with average population of 10^4^ cells were cultured overnight in each well of 96-well plates, using DMEM (High Glucose) and kept in incubator at 37 °C, with 5% CO_2_. A hundred microliter of each decuple (i.e., ten-fold) dilution steps was transferred to each well of the 96-well plates. Supernatant of the infected cells was collected after 48–96 h, and cytopathic effect (CPE) of the cells was monitored using optical microscope. Viral titration was calculated using the Spearman-Karber method [6]. The median tissue culture infectious dose (TCID50) of the selected strain was calculated using the Reed–Muench method [7]. Genomic content of Vero cells of each well with minimal number of plaques was extracted for further molecular characterization. This process was repeated until a single-plaque was obtained and its molecular identity was approved. At the end, the isolated strain was considered as the FAKHRAVAC vaccine seed, which was aliquoted and preserved in −80 °C. During quality control of the candidate vaccine’s production, each batch of formulated vaccine that was injected into either of the animal models was first tested for not having a live virus. The samples that acquired from each batch of produced vaccine were transferred to the Vero-cell culture media and monitored for four to five days to make sure that no plaque formation or CPE occurred.

### 2.4. Vaccine Preparation and Identification

All stages of vaccine preparation are represented in a diagram (Figure 1A). First, Vero E6 cells were cultured in 12-well plates and then infected by SARS-CoV-2. Second, cell cultures supernatant recovered and clarified by centrifugation (Centrifugation at 1000× *g* for 10 min at 4 °C) and microfiltration (Polypropylene filter, Polygard CN 0.3 µm; NFF), respectively. Optimal titer of the inactivated virus was formulated with Al(OH)_3_. The process of concentration and final purification of the inactivated virus was continued. To this end, 250 milliliters of the supernatant were filtrated via Sartorius cassette filter (50 kDa cut-off) using a peristaltic pump, and condensed to a volume of 10 milliliters. Third, inactivation of the virus was performed by formaldehyde addition (1% *v*/*v*, Merck, Darmstadt, Germany). Formaldehyde (37% pharmaceutical grade) added to the suspension dropwise (1 mL/min). The final concentration of formaldehyde in the suspension was 1% (*v*/*v*), and the suspension stirred gently for 24 h at room temperature (22–25 °C). The fourth and fifth, inactivated virus was ultra-filtrated via ÄKTA pure FPLC (Fast Protein Liquid Chromatography) system via one step chromatography by use of specific resin in ÄKTA system. Sixth, assessment of the purification/condensation process was monitored using TCID50 (median tissue culture infectious dose)-targeted ELISA (enzyme-linked immunosorbent assay). Identification and assessment of purified vaccine was carried out by transmission electron microscopy (TEM), SDS-PAGE (polyacrylamide gel) and western blot methods (Figure 1B). The structure of the isolated virus was monitored using the transmission electron microscope (TEM, EM208S, Philips, at final magnifications of 9 × 10^4^). Inactivated virus suspension after being concentrated with ultrafiltration, was run in SDS-PAGE with 10% polyacrylamide and 2-mercaptoethanol. The extracted antibodies, produced by rabbit after wild virus administration, were employed in the western blot analysis.

### 2.5. qRT-PCR

Total RNA was extracted from samples using RNeasy Mini Kit (QIAGEN, Hilden, Germany). The SARS-CoV-2 nucleocapsid protein (N) and RNA-dependent RNA-polymerase (RdRp) genes were detected using SENMURV multi-star SARS-CoV-2 RT-PCR kit (STRC, Iran). The qRT-PCR program was performed under the following reaction conditions: 50 °C for 20 min, 95 °C for 10 min, and 40 cycles of 95 °C and 55 °C for 10 and 40 s, respectively.

### 2.6. Humoral Immunization

In order to obtain minimal antigen content in FAKHRAVAC formulation, which results in maximal antibody production, two approaches of injection were taken into account. In the first approach, Balb/c mice and Guinea pig received vaccine candidate intraperitoneally and intramuscularly, respectively, via double/triple infusion of three different concentrations of high (H): 10 μg/dose, medium (M): 5 μg/dose, and low (L): 1 μg/dose. In the second approach, all of the rabbits and RMs received (M, H) and H dosages of FAKHRAVAC via intramuscular infusion, respectively.

All animals were randomly divided into four groups. For immunization, viral preparations were emulsified with aluminum hydroxide adjuvant in a way that three groups of H, M and L were established. Further, a control group that received aluminum hydroxide alone was considered as placebo. Each group of animals involved Balb/c mice (Either of H, M, and L doses; n = 76, placebo (negative control: C-); n = 78), guinea pig (either of H, M, and L doses; n = 54, placebo (C-); n = 54), rabbits (either of H, M, and L doses, placebo (C-); n = 33), and RMs (either of H, M, and L doses; n = 5, placebo (C-; n = 3).

Blood samples were collected from each animal model before immunization, and the serum was isolated the next day as control [3]. Following double (day 0, 14: 0/14 and day 0, 21: 0/21) and triple (day 0, 14, 21: 0/14/21) infusions, sera were diluted in ratios of 132^−1^ (1//132), 528^−1^ (1//528) and 2112^−1^ (1//2112), and their optical density at 450 nm was measured.

The titer of the IgM and IgG antibodies generated against viral antigens of inactivated virus and then specified by indirect ELISA. The 96-well plates were coated with isolates of viral strain. After blocking with 5% skim milk, the plates were subsequently washed and incubated with sera (according to diluted amounts). They were then washed with Phosphate-Buffered Saline–Tween (PBS-T) and incubated for one hour following the addition of alkaline phosphatase conjugated rabbit anti-mice IgG (Promega, Madison, WI, USA). After incubation for 10 min, the reaction was stopped by the addition of 2.5 M H_2_SO_4_. The absorbance was measured at 492 nm using a microplate reader [8].

In 0/14 days vaccination program, the serum was isolated the next week after the first injection, before the second injection, and one, two, and three weeks after the second injection. In 0/21 mode, the serum was isolated in one and two weeks after the first injection, before the second injection, and one and two weeks after the second infusion. In 0/14/21 mode, the serum was isolated before the second and third injection, and one, two, and three weeks after the third injection. Each animal was inoculated with 0.5 mL of the test sample (equivalent to 1 human dose) according to the method proposed by Wang et al. [3].

### 2.7. Vaccine Immunogenicity Analysis and Protection in Animals

#### 2.7.1. Neutralization Antibody (NAb) Assay

To assess the immunogenicity of FAKHRAVAC, BALB/c mice were injected with different immunization programs and various doses, and levels of NAb were evaluated at 0, 14 and 21 days after injection. The seroconversion rate in the H, M and L dose groups was measured in Balb/c mice and guinea pig via double (0/14, 0/21) and triple (0/14/21) administration. Experimental design of NAb titration in dose group and injection order of rabbit and RM was carried out according to the results that were obtained by NAb titration of Balb/c mice and guinea pig. Blood samples, obtained from immunized animals, centrifuged at 6000 rpm for 10 min at 4 °C, to obtain serum, which was inactivated at 56 °C for 30 min. The inactivated sera were diluted by 10, 100, and 1000 times, and supplemented with suspension of active virus (600 μg/dose) by 1:1 ratio at final volume of 1 ml for two hours. Then, the 1:1 suspension was added to the cell culture media (cultured Vero cell with confluency >70%) and kept in CO_2_ incubator for four days at 36 °C. Then, NAb titration, emergence of CPE, and plaque formation was monitored at 36 °C for 7–10 days. The supernatant of the cell culture was collected via “5-min” centrifugation at 4000 RCF, FBS free, and decuple serial dilution of the supernatant was prepared at 36 °C, to infect E6 Vero cells that were already cultured in 12-well plates, respectively. Incubation of Vero cells with viral dilution was extended until plaque formation. By observing the CPE, the neutralization endpoint was calculated considering both serum dilution and ratio of CPE formation in cell cultures. The Karber method was used to calculate the neutralization endpoint (the serum dilution in logarithmic scale) [9]. Then, the highest dilution of serum that protects 50% of cells from infection, compared to the 100 TCID_50_ is considered as the antibody potency of the serum. A neutralization antibody potency is considered as negative when <1:2, while that R 1:2 is positive.

#### 2.7.2. Challenge Assay in Rhesus Macaques

Two RMs received active SARS-CoV-2 with concentration of 300 μg/dose using a nasal spray (n = 2) under anesthesia followed by 0/14 immunization program, while RMs of placebo group (n = 2) received no vaccine. Virus challenge started in both nasal spray and placebo groups and the viral load monitored by sample acquisition via oral and anal swabs and qRT-PCR analysis up to seven weeks of the time-course day. Measurement of RMs’ body temperature and weight, alongside peripheral blood collection was carried out to monitor fever and IgG titration, in both groups from 0 to 50 days post inoculation (dpi) [2]. After seven weeks, all animals were euthanized and a pathological examination was conducted.

### 2.8. Safety Evaluation

Eighteen Balb/c mice of both genders were divided in three groups (n = 6), beside a control group (n = 2). They were injected intramuscularly with H, M, and L dosage of FAKHRAVAC vaccine and aluminum hydroxide as placebo, respectively. Four rabbits were divided into three groups and injected with sodium chloride (negative control), M and H doses of FAKHRAVAC. Three RMs were divided into two groups and intramuscularly injected with a sodium chloride (negative control) and H dose of FAKHRAVAC in a volume of 0.5 mL. The animals were injected in double (0/17, 0/28 days) and triple (0/17/28 days) infusion modes.

Clinical manifestation, body weight and body temperature of the animals were monitored during and after immunization. Lymphocyte subset percentages (CD4+, CD8+ and CD4+/CD8+ ratio) of the collected blood samples were calculated. Samples were prepared and acquired according to instructions of an Attune Acoustic Focusing Cytometer (Life Technologies) and analyzed using FlowJo software.

### 2.9. Phylogenic Tree Analysis of SARS-CoV-2 Virus

Eleven genome sequences for SARS-CoV-2 that acquired from Iranian patients were retrieved from NCBI (https://www.ncbi.nlm.nih.gov/nuccore/ (accessed on 21 September 2021)). Besides, 25 of the latest registered GeneBank sequences of SARS-CoV-2 complete genome, from countries with highest total death in five continents, were clustered by MUSCLE algorithm (UPGMA cluster method), in addition to the indigenous strain that we used as viral material for vaccine (FAKHRAVAC). This selection was also based on haplotypes that were identified previously [10], to analyze geographic and chronologic distribution of the S-protein. Further, a phylogenetic construction was performed by maximum parsimony tree, using MEGAX software [11]. Members of the clade(s) that had similar bootstrap score have been analyzed using spike protein sequences protein, using ClustalW algorithm [12].

Tertiary structure of the FAKHRAVAC and Wuhan S4 protein was retrieved via protein data bank (7CWM) and homology modelling (SWISS-MODEL Workspace), respectively [13]. The structural format of the proteins was then analyzed by Molegro virtual docker as what is explained in previous studies [14,15].

### 2.10. Statistical Analysis

The Prism software Version 6 used to analyze the level of significance, using one-way ANOVA and Bonferroni post-test methods.

## 3. Results

In the following sections we show how the isolated viral strain resulted in humoral immunization in animal models, while did not come up with elevation of inflammatory markers.

### 3.1. Vaccine Design and Production

In our production procedure, ‘high virus replication titers (2 × 10^8^ CCID50) were obtained’ after 24–72 h post-infection. Inactivation of virus by formaldehyde continued up to four batches, and resulted in complete inactivation (Figure 1A). The SDS-PAGE and western blot analysis by use of COVID-19 patient sera of purified vaccine demonstrated virus structural proteins (Figure 1B). Transmission electron micrograph of our candidate vaccine revealed an intact spherical viral structure with crown spikes and almost 100 nm in diameter (Figure 1B).

The phylogenetic tree made out of the complete sequences of the SARS-CoV-2 strains, suggests four distinct clades that are categorized as clades one to four (Figure 1C). The strain of the viral seed that used in our study “FAKHRAVAC” is categorized in clade one, along with the most recent strain of SARS-CoV-2 reported from Australia and UK “P. Britain”, while the majority of the other viral strains that were reported from Iran were categorized in clade two along with isolates from Iraq, Wuhan, and Spain. The complete genome of the lineages B.1.1.7 (Alpha variant), B.1.351 (Beta variant) and B.1.617.2 (Kappa variant) has been retrieved from NCBI-VIRUS and indicated with a “P” at the beginning of their name P.Britain, P.South-Africa, and P.India.

The RBD (Receptor Binding Domain) sequence of the spike protein “S” from the strains of clade one, were retrieved from NCBI-Virus database and aligned using ClustalW algorithm [16]. The S-protein’s RBD of the Wuhan isolate was considered as reference (Figure 1D). The RBD is known to play a major role in interaction with Angiotensin-converting enzyme 2 (ACE2) that is attached to the membrane of human lung, artery, heart, kidney, and intestine cells [3]. Since ACE2 serves as the entry point into cells for SARS-CoV-2, any mutation that might result in structural reconfiguration of the S-protein, especially RBD region, is supposed to have an impact on the pathogenicity of the corresponding strain [17]. There are three substitutions; D138 → Y13 → 8, S477 → N477, and D614 → G614, when S-protein of the Wuhan and FAKHRAVAC were compared. The S477 → N477 substitution is placed in RBD region (Figure 1D). Only the S-protein of FAKHRAVAC and isolated strain from Australia obtained such a kind of substitution, while other members of clade one still had S477 in their RBD region. None of the three substitutions had resulted in secondary structure variation. However, FAKHRAVAC S-protein tertiary structure was different from its Wuhan homolog, which was assumed to be correlated with D138 → Y138 and D614 → G614 substitutions. In recent substitutions, the negatively charged aspartic-acid was replaced by either hydrophobic Tyrosine (Y138) or Glycine (G614), and the FAKHRAVAC-S-protein got less tightly packed, compared to its Wuhan homolog (Figure 1E,F).

### 3.2. Immunogenicity in Mice, Guinea Pig, Rabbit and RMs

To assess the humoral immunogenicity of FAKHRAVAC, BALB/c mice, guinea pig, rabbit, and RM were injected with double (0/14 and 0/21 days) and triple (0/14/21) immunization programs (Figure 2A,B). The BALB/c mice and Guinea pig received high (H), medium (M) and low (L) doses of the candidate vaccine, while rabbits received “M and H” and RMs received H doses. The level of neutralization antibody (NAb) was evaluated after injection in each group. The IgG titration, as indication of humoral immunogenicity, was monitored by ELISA measurement.

The IgG titration results acquired from BALB/c mice and Guinea pig showed that in triple infusion mode (0/14/21), the IgG boosted considerably in two steps by days 14 and 28. In other words, IgG considerably elevated on the 14th and 28th day compared to its previous time course checkpoint (i.e., day 0 and 21), respectively, which is called a “boost” from now on in this study. It must be noted that the highest IgG titration obtained in mice on the 21st day after vaccination. The IgG titration remained the same by 42 days, when either of 1//132, 1//528, or 1//2112 serum dilutions were tested (Figure 2A,B).

When mice received double injections (0/14 and 0/21), first IgG boost was obtained after one week. The second IgG boost observed in day 21 and 28, after the time that the second shots administered in 14th and 21st days in 0/14 and 0/21 modes, respectively (Figure 2A).

When guinea pigs received double infusions (0/14 and 0/21), usually three IgG boosts were obtained, except for the time that medium dosage was administrated in 0/14 mode (Figure 2B). The first boost always obtained after one week, while second and third boosts obtained on the 28th and 42nd days (in 0/21 mode—Figure 2B). However, the second IgG boost obtained in 21st day in 0/14 mode, while the third boost obtained in day 21, 28, and 35 of 0/14 mode, when medium, high, and low dosages of vaccine candidate were injected, respectively (Figure 2B).

The IgG titration results acquired from rabbits showed that IgG always boosted in two steps either by days 7 or 21 (in 0/14 and 0/21 modes) or by days 14 and 21 and 14 and 28 when medium and high dosages of the candidate vaccine was administrated in 0/14/21 mode, respectively (Figure 3A). There are two boosts of IgG titration in the RM, as well as for rabbits. However, when high dosage of the candidate vaccine was injected to the RMs via 0/21 mode, five boosts were observed on days 7, 14, 21, 28, and 42 (Figure 3B-left panel). In other word, IgG level on the 7th day was considerably higher than day 0, and as the same pattern repeated in consecutive checkpoints of time-course. Therefore, the IgG level in day 7, 14, 21, 28, and 42 was considerably higher than day 0, 7, 14, 21, and 28, respectively, which eventually made a five-boost pattern (Figure 3B-left panel).

The levels of neutralization antibody (NAb) were evaluated in 0/14, 0/21 and 0/14/21 immunization programs. The seroconversion rate at the high, medium, and low dose groups of all tested animals reached 100% by 28 days after immunization (Figure 4). Further, variation of the NAb levels differed by immunization program, in a way that the 0/14/21 regimen scored the highest NAb level in tested RMs.

### 3.3. Safety

In this study, 10 Balb/C mice were divided into two groups of control and treatment. The mice in treatment group subcutaneously injected with one dose (Triple High = 30 μg) of candidate vaccine and physiological saline as the control. All mice were monitored for 48 h to assure that there is no case of death, significant difference in weight or feeding state, and histopathologic changes after euthanasia (15 days after vaccination). It should be noted that the maximum tolerated dose (MTD) used for a “triple high” shot in mice, is hundred times more than the dose in humans, which makes it a reliable candidate vaccine as a matter of safety. Monitoring lymphocyte subgroup distribution (CD4+, CD8+ and the CD4+/CD8+ ratio) in all tested animals (Balb/c mice, Rabbit and RMs) showed that neither CD4+, nor the CD8+ and CD4+/CD8+ ratio, differed after vaccination. This finding was irrespective of either immunization program or administrated dosage (Figure 5).

### 3.4. Protection in a Non-Human Primate Animal Model

All RMs in challenge group were immunized twice on days 0 and 14, while the RMs in placebo group only received intramuscular placebo on those days. The viral load that was measured by real-time PCR, performed via oral and anal sample acquisition from RMs, which confirmed higher viral load in placebo group than the challenge group. IgG titration, alongside monitoring of RMs’ body temperature and weight, was carried out before the virus challenge assay, and was repeated weekly up to 42 days after challenge (Figure 6A–D). The IgG titration showed a considerable elevation in two weeks after infection in challenge group, which remained increased up to 42 days after vaccination. However, statistical analysis of RMs’ body temperature and weight showed that there is no considerable difference between monitored groups.

SARS-CoV-2-infected RMs in placebo group developed significant pulmonary infiltrates and histological lesions compared to the challenge group, which is represented by increased number of red blood cells in alveolar walls and elevated presence of inflammatory factors in interstitial tissue (Figure 6E). Further, the lung histopathological sections that were stained with hematoxylin and eosin (H&E), demonstrated increased thickness of bronchioles in the placebo group compared to the challenge group. Histological evaluation performed by three different pathologists, using comprehensive method of histopathologic examination [18]. The histopathologic indices that were taken into account, scores to the extent of epithelial thickening, extracellular discharge and alveolar inflammation (Figure 6F). The interpretations of data measurements were reported as ordinal (i.e., normal = 0, mild = 1, moderate = 2, and severe = 3). Monitoring the body temperature of RMs in challenge group conferred highly efficient protection against SARS-CoV-2 in RMs that received FAKHRAVAC with no considerable ADE symptoms.

## 4. Discussion

Development of safe immunogenic vaccines is crucial for control of the global COVID-19 pandemici. Here, we developed an inactivated SARS-CoV-2 vaccine candidate, FAKHRAVAC, which induced immunization in four mammalian species with neither ADE nor significant side effects.

Results of FAKHRAVAC humoral immunogenicity in four mammalian animal model showed that the maximal IgG boost observed by day 14 when H-dosage of FAKHRAVAC was injected to Balb/c mice and Rabbit, while, the highest IgG boost in Guinea pig and RM obtained by day 21 when H dosage of FAKHRAVAC were administrated. These finding suggest that the more the animal weight got, the later IgG boost obtained at the expense of H-dosage of candidate vaccine. The seroconversion rate in the H-dosage reached to maximum in the 0/14/21 regimen, which confirms the immunization effect is time dependent in all animals. Further, investigation of the FAKHRAVAC safety in all mammalian animal models demonstrated that the level of inflammatory markers (CD4+, CD8+ and the CD4+/CD8+ ratio) remained the same after vaccination.

Previous studies conferred *Rhesus macaques* (RMs) that were infected with SARS-CoV-2, developed pulmonary infiltrates and histological lesions. That is why they are suitable candidates for modeling SARS-CoV-2 infection [19]. In this study, the challenge assay of SARS-CoV-2 conferred highly efficient protection against SARS-CoV-2 in RMs that received FAKHRAVAC, which remained considerably effective until the end of monitoring time-course.

Assessment of FAKHRAVAC immunization potential in pre-clinical study state introduces a promising vaccine candidate, which is comparatively effective as its premier inactivated vaccines [2,3]. However, one of the major concerns about the vaccine development is its potential of immunization against new SARS-CoV-2 lineages that are breaking out in Britain, South Africa, and India. To know more about this challenge, complete genome of the lineages alpha, beta, and kappa has been retrieved from NCBI-VIRUS that were shown up with a “P” at the beginning of their name, which stands for prevalent. Lineage alpha as a variant of SARS-CoV-2 is estimated to be more transmissible than the wild-type SARS-CoV-2. This increase is thought to be at least partly due to one or more mutations in the virus’s spike protein. The variant is also notable for having more mutations than normally seen [20]. There are three mutations in the spike region of the beta lineage genome (K417N, E484K, N501Y) [21]. The N501Y mutation is found in both of alpha and beta lineages, which does not result in structural variation of the S-protein, compared to the Wuhan reference. The phylogenetic analysis of SARS-CoV-2 complete genome showed that the FAKHRAVAC stands in clade one with strains isolated from Australia, UK (regular and alpha lineage), Nigeria, Egypt, and Russia. Lineage kappa that is known as an out-breaking variant of SARS-CoV-2 in India, has been referred as double mutation variant [22], which is categorized in clade four along with strains isolated from India, Italy, South Africa, Iran, and Tunis. On 7 May 2021, the delta lineage (B.1.617.2) has been flagged as a variant of concern (VOC), based on an assessment of transmissibility, made by Public Health England (PHE) [23]. In the list of countries reporting variants of concern as of 9 March 2021, reported by WHO, the prevalence of VOC 202012/01 (B.1.1.7-alpha lineage) in Iran was verified [24].

## 5. Conclusions

Currently, a Phase III clinical trial of FAKHRAVAC is in progress (IRCTID: IRCT20210206050259N3), using the same aluminum adjuvant formulation. According to the current preclinical study, the complete genome of FAKHRAVAC viral seed, contrary to other isolated strains from Iran, is more akin to recently mutated variants of SARS-CoV-2 (e.g., alpha lineage). Further, the configurational variations of predicted tertiary structure of FAKHRAVAC’s viral seed S-protein, compared to Wuhan homolog, provides a promising fact that FAKHRAVAC is expected to immunize recipients against newly reported B.1.1.7 lineage as well as regular ones.

## Figures and Tables

**Figure 1 vaccines-09-01271-f001:**
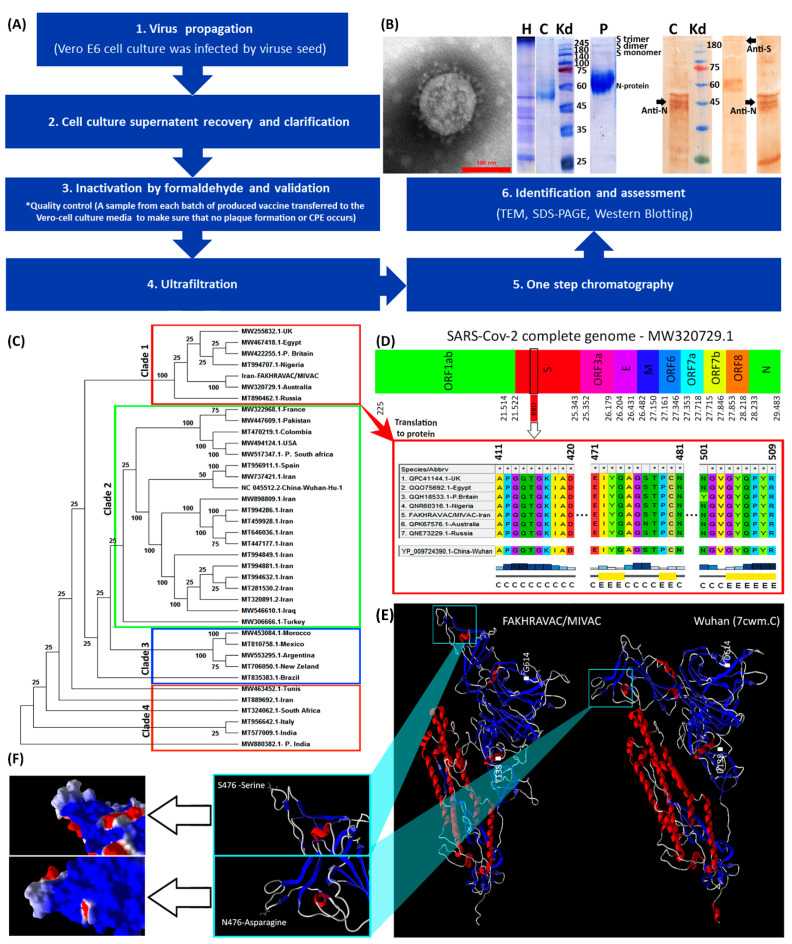
Characterization of viral strains. (**A**) A summary diagram of the procedures that were involved with vaccine production. (**B**) From left to right: TEM, SDS-PAGE, and western blot results. The protein composition of FAKHRAVAC was evaluated by incubating with antibodies targeting N and S protein, which are addressed in the figure by black arrows. The topics on the lanes represents: H, harvest, C, Serum of convalescent patient; and P, purified viral solution. (**C**) Phylogenetic construct made out of the complete sequence of the SARS-CoV-2 *m-RNA*. (**D**) Infographic view of the coding regions of the SARS-CoV-2 virus with emphasis on multiple-alignment of their RBD regions. The letters “C” and “E” under the multiple-alignment represents secondary structure elements of “Random-Coil” and “Beta sheet” elements, respectively. (**E**) Tertiary structure of the FAKHRAVAC and Wuhan S-proteins, with a zoomed-in image on the RBD region. (**F**) Electrostatic surface of the S-protein’s RBD structure. The red, white, and blue represent negative, neutral, and positive charge, respectively.

**Figure 2 vaccines-09-01271-f002:**
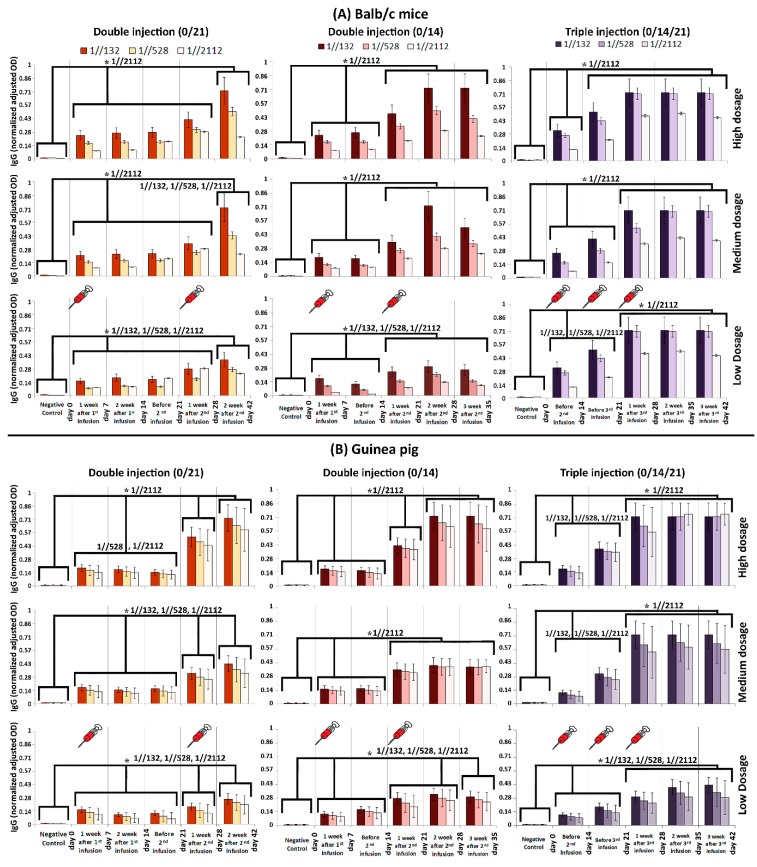
Humoral immunity in (**A**) Balb/c mice and (**B**) guinea pig monitored via IgG titration. Three dilutions (1//132, 1//528 and 1//2112) were prepared for each of the acquired sera samples. The humoral response was measured after double and triple infusions. The horizontal axis in each graph represents the time line of monitoring. The dosage of administrated vaccine candidate is depicted in the right of the figure. The comb-like bars above the columns of the graphs represent the groups of results that came out with significant differences in IgG titration (ELISA). The star above the comb represents which sera dilutions of comparing categories are different. For example, as what is shown in the top left panel, the IgG (OD) of “Negative control” is considerably less than the interval of “day 0–28”, while the IgG (OD) at “day 42” is considerably higher than both of “Negative control” and “day 0–28” interval, when the sera was diluted by 1//2112-fold. The vertical axis represents the normalized adjusted optical density (OD) of IgG at 450 nm. The obtained sera were diluted by 132, 528, and 2112 folds, and in cases that the recorded OD was more than one, the samples were diluted again, in a way that their absorbance gets ≤1. Then, the dilution rate (until the absorbance got ≤1) was multiplied by the OD to obtain an adjusted OD. In order to have a uniform and comparable range of adjusted ODs in all graphs, they were normalized in the range of 0–1. *p*-values of less than 0.05 were set as a cut-off for statistical significance.

**Figure 3 vaccines-09-01271-f003:**
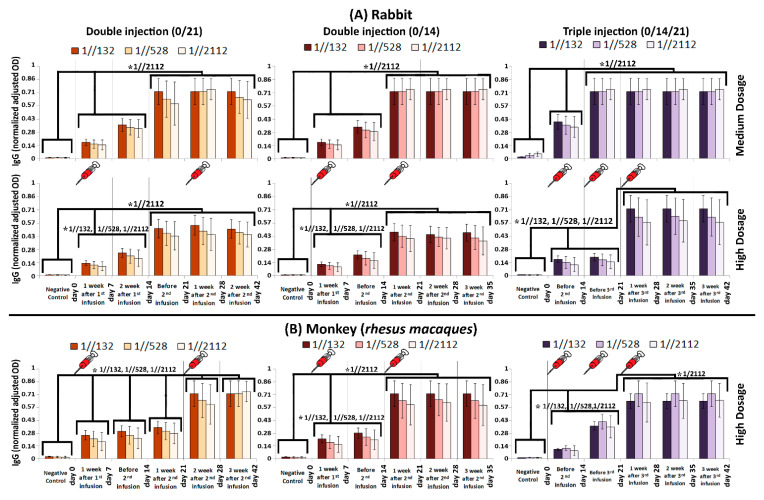
Humoral immunity in (**A**) Rabbit and (**B**) RM monitored via IgG titration. In case that we consider sera from infected RMs as “positive control”, the IgG (normalized adjusted OD) after 132, 528, and 2112 folds of dilution would be 1.9 ± 0.3, 2.1 ± 0.4, and 1.8 ± 0.6, respectively. The comb-like bars above the columns of the graphs represent the groups of results that came out with significant differences in IgG titration, while the schematic syringes represent days that injections were carried out. The star above the comb represents which sera dilutions of comparing categories are different. For example, as what is shown in the top left panel, the IgG (normalized adjusted OD) of “Negative control” is considerably less than the interval of “day 0–14”, while the IgG (normalized adjusted OD) at “day 21–42” interval is considerably higher than both of “Negative control” and “day 0–14” interval, when the sera was diluted by 1//2112 folds. The vertical axis represents normalized adjusted OD, which is explained in Figure 2 caption. *p*-values of less than 0.05 were set as a cut-off for statistical significance.

**Figure 4 vaccines-09-01271-f004:**
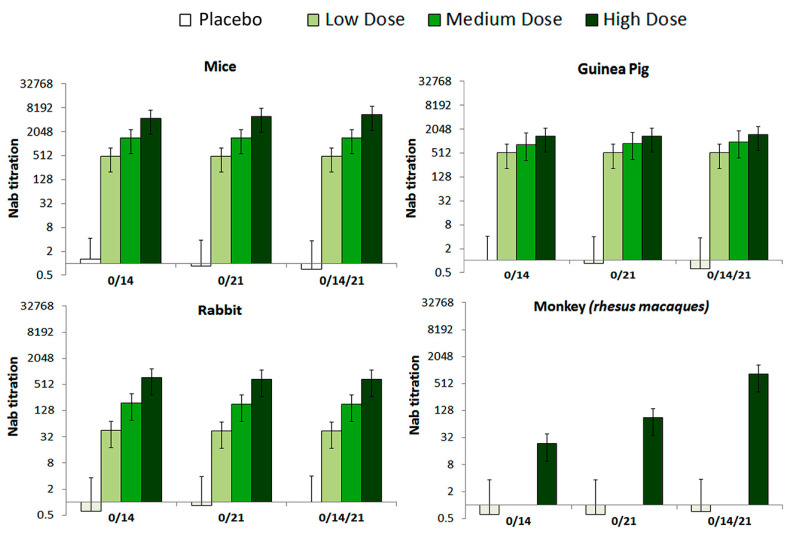
The NAb levels of mice, Guinea pig, Rabbit and *Rhesus macaques* monkey with different immunization programs. Animals were injected with High (10 μg), Medium (5 μg), or low (1 μg) doses of vaccine, correspondingly via, two-dose (0/14, 0/21), and three-dose (0/14/21) immunization programs, respectively. The NAb levels measured by microtitration method, 28–42 days after the first immunization (n = 10 for all animals). The serum dilution is calculated according to Karber method in logarithmic scale (Log2(1:X)) and what is presented in graphs is the Nab titration, which is result of 1:10, 1:100 and 1:1000 dilution of active virus/sera.

**Figure 5 vaccines-09-01271-f005:**
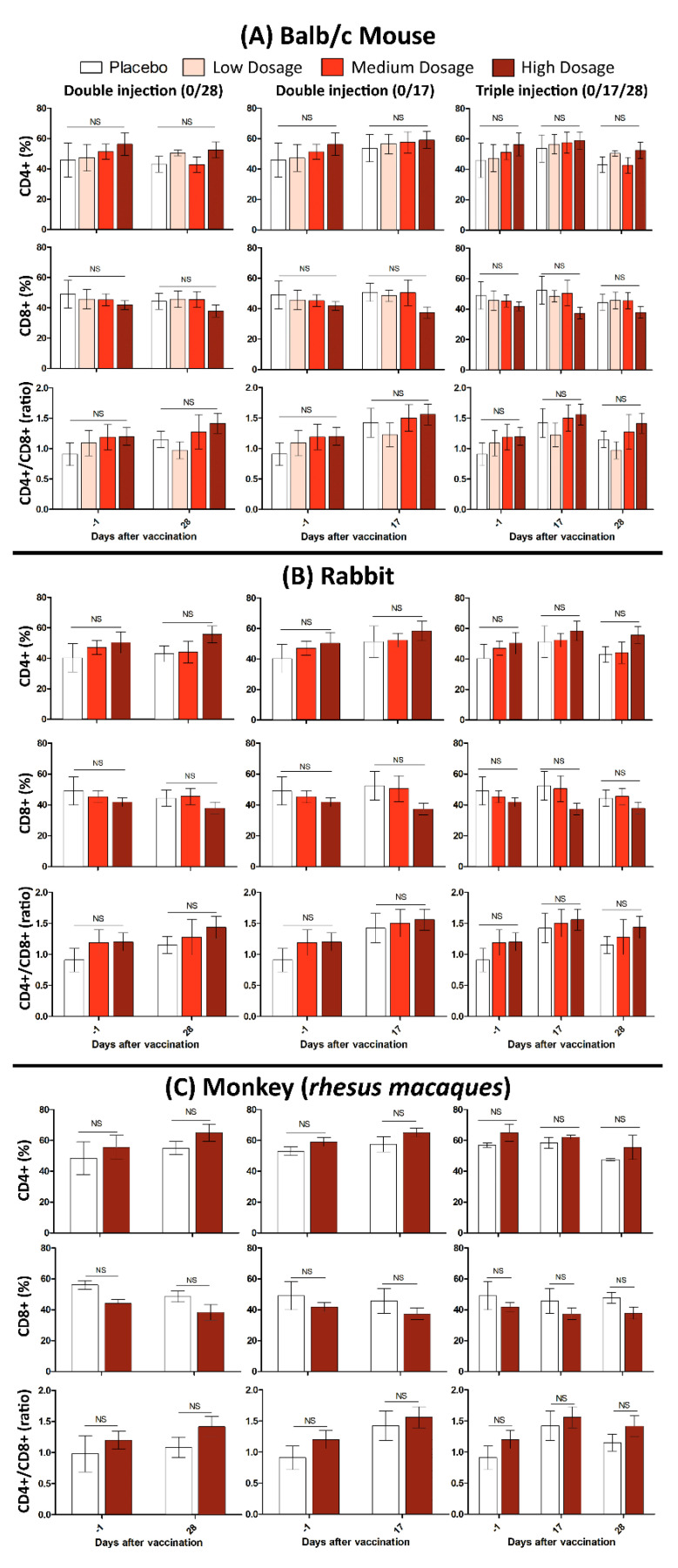
Cellular immunity evaluation after administration of low, medium, and high dosages of candidate vaccines in (**A**) Balb/c mice, (**B**) Rabbits, and (**C**) RMs. The error bars represent standard deviation of repeated experiments and “NS” at the top of bars indicates a non-significant difference between the averaged experiments.

**Figure 6 vaccines-09-01271-f006:**
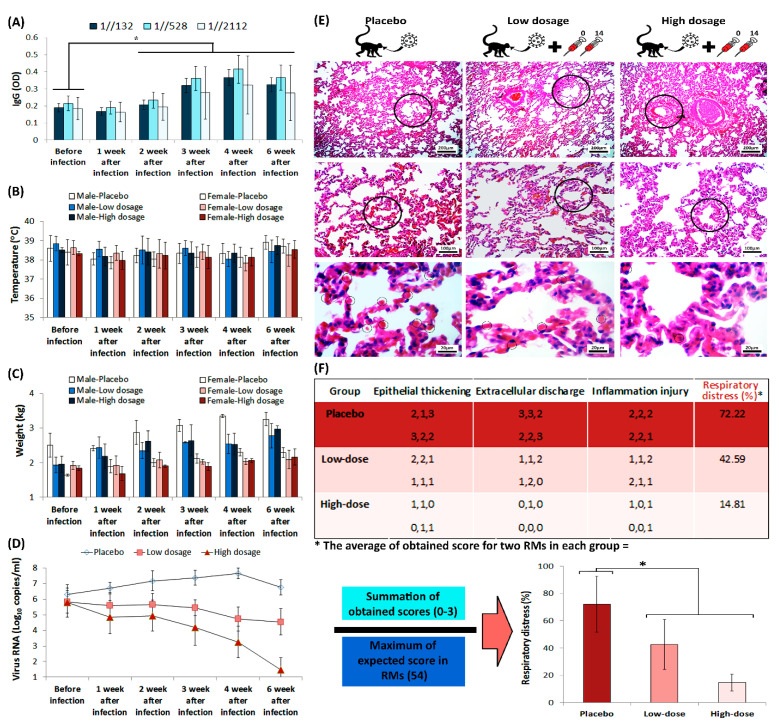
The humoral immunity of RMs during challenge assay. (**A**) IgG titration during challenge assay. The star above the bars represents the significant difference, compared to the control state (before injection) and the 1//132, 1//528, and 1//2112 represent sera dilutions. The “y axis” represents optical density of IgG at 450 nm. (**B**) Monitoring of both male and female RMs’ body temperature, and (**C**) weight, during challenge assay. (**D**) Viral loads in RMs, acquired from throat and anal swabs, were measured by real-time PCR. (**E**) RMs’ lung H&E staining, prepared from placebo and challenge group animals after euthanasia, seven weeks after infection. The circles in the top, middle and low rows represent bronchiole, alveoli and red blood cell, respectively. (**F**) Scoring of the histopathological images is represented as a matter of respiratory distress percentage (%). The star above bars represents significant difference between series (*p*-values of less than 0.05 were set as a cut-off for statistical significance).

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
