# Peer review of "Development of Inactivated FAKHRAVAC® Vaccine against SARS-CoV-2 Virus: Preclinical Study in Animal Models"

_vaccines, 2021, doi:10.3390/vaccines9111271_

Round 1

Reviewer 1 Report

The manuscript is testing the efficacy of an inactivated SARS-CoV-2 vaccine. Overall, it appears that the vaccine induces an antibody response, although it is not clear if this is protective as sufficient data is not provided to determine this. It was also difficult to understand some of the methods and results. Specific concerns are indicated below:

  • For the challenge experiment, they do not show the viral load data. I was not able to open the supplementary file, so if it was in there, I did not see it.
  • To demonstrate protection, they could have plotted the temperature and weight over time.
  • They did not indicate what time point the lung pathology is from. In order to determine whether there is a difference between the groups, sections from multiple animals need to be scored in a blind manner.
  • It is not clear how the neutralization assay was performed. Based on reading the methods, it seems like a two-step process was used, which is not conventional. In addition, negative and positive controls are not shown for this assay.
  • For the ELISA data in Figures 2 and 3, it is not clear what comparisons are being made that are indicated as significant.
  • For the ELISA data, positive controls of sera from infected individuals would be helpful to know how the level of antibody after vaccination compares.
  • The y axis on the ELISA data is labeled as IgG titration (OD), but the unit is just the OD. It should read IgG (OD).
  • In many of the ELISAs, the serum is not titrated sufficiently.
  • On lines 302-304, they indicate that there were five boosts in antibody levels in the RMs, however, this is not evident in the data. It appears there is only 1 boost in antibody level.
  • Lines 312-313 state that the 0/14/21 regimen scored the highest Nab level in all tested animals, but the graphs for the different groups look identical.

Author Response

Comments and Suggestions for Authors

The manuscript is testing the efficacy of an inactivated SARS-CoV-2 vaccine. Overall, it appears that the vaccine induces an antibody response, although it is not clear if this is protective as sufficient data is not provided to determine this. It was also difficult to understand some of the methods and results. Specific concerns are indicated below:

Author reply: Dear reviewer, thanks for the time that you put through revision of the manuscript. Your points about the “protection” and “difficulty of understanding” are taken. We tried to address your concerns through following replies. 

  1. For the challenge experiment, they do not show the viral load data. I was not able to open the supplementary file, so if it was in there, I did not see it.

Author reply: That is right. We added viral load data in section 2.7.1 as follows:

“The inactivated sera was diluted by 10, 100 and 1000 times, and supplemented with suspension of active virus (600 μg/dose)”

“Two RMs received active SARS-Cov-2 with concentration of 300 μg/dose using a nasal spray (n=2)”

  1. To demonstrate protection, they could have plotted the temperature and weight over time.

Author reply: We have added monitoring of the temperature and weight over time in Figure 6. The sentence in section 3.4 is rewritten as follow:

“IgG titration, alongside monitoring of RMs’ body temperature and weight, was done before virus challenge assay, and was repeated weekly up to 42 days after challenge (Figure 6A-C).”

The figure-6 caption is updated as follows:

“(B) Monitoring of both male and female RMs’ body temperature, and (C) weight, during challenge assay.”  

  1. They did not indicate what time point the lung pathology is from. In order to determine whether there is a difference between the groups, sections from multiple animals need to be scored in a blind manner.

Author reply: The time point of the lung pathology is explained in Figure-6 caption:

“(D) RMs’ lung H&E staining, prepared from placebo and challenge group animals after euthanasia, seven weeks after infection.”

What is shown in Figure-6D represents a schematic difference between the groups. Further, histopathologic images of multiple RMs scored as a matter of acute respiratory distress in a blind manner. Here are the extra details:  

“Histological evaluation performed by three different pathologists, using comprehensive method of histopathologic examination [19]. The histopathologic indices that were taken into account, scores to the extent of epithelial thickening, extracellular discharge and al-veolar inflammation (Figure 6E). The interpretations of data measurements were reported as ordinal (i.e. normal= 0, mild= 1, moderate= 2 and severe= 3).”

Besides, a new section is added as “Figure-6D”

(E) Scoring of the histopathological images is represented as a matter of respiratory distress percentage (%). The star above bars represents significant difference between series (P-values of less than 0.05 were set as a cut-off for statistical significance).

  1. It is not clear how the neutralization assay was performed. Based on reading the methods, it seems like a two-step process was used, which is not conventional.

Author reply: We merged two parts of “Neutralization antibody (NAb) assay” and “Challenge assay” as sub-sections of “Vaccine immunogenicity analysis and protection in animals”.

According to section 2.7.1, our neutralization assay is expressed in following steps:

“1-Animals were injected with vaccine via different immunization programs and various doses

2- The seroconversion rate “virus vs serum titration” was done.

3- The 1:1 suspension was added to the cell culture media.

4- Then, emergence of CPE and plaque formation monitored for 7-10 days.

5- The Karber method was used to calculate the neutralization endpoint.

For example, the average Nab titration in case of H-dose injection to RM at 0/14/21 day would be calculated as follows: =2^(-(LOG(0.001,2)-(LOG(1,2)/2)+9/9))= 824.84 ± 247.45, which could be found in figure 4.

The original CPE table that the results are extracted from is as follows:

Animal and serum acquisition day

Virus/serum dilution ratio

Dosage and days of injection

Virus 1/10

Serum1/10

Virus 1/10

Serum1/100

Virus 1/10

Serum1/1000

Virus 1/100

Serum1/10

Virus 1/100

Serum1/100

Virus 1/100

Serum1/1000

Virus 1/1000

Serum1/10

Virus 1/1000

Serum1/100

Virus 1/1000

Serum1/1000

BALB/C Mice, D28

H, 0/14

CPE Observed

CPE Observed

CPE Observed

No CPE Observed

CPE Observed

CPE Observed

No CPE Observed

CPE Observed

CPE Observed

BALB/C Mice, D42

H, 0/21

CPE Observed

CPE Observed

CPE Observed

No CPE Observed

CPE Observed

CPE Observed

No CPE Observed

No CPE Observed

CPE Observed

BALB/C Mice, D42

H, 0/14/21

No CPE Observed

CPE Observed

CPE Observed

No CPE Observed

CPE Observed

CPE Observed

No CPE Observed

No CPE Observed

CPE Observed

BALB/C Mice, D28

M, 0/14

CPE Observed

CPE Observed

CPE Observed

CPE Observed

CPE Observed

CPE Observed

CPE Observed

CPE Observed

CPE Observed

BALB/C Mice, D42

M, 0/21

CPE Observed

CPE Observed

CPE Observed

CPE Observed

CPE Observed

CPE Observed

CPE Observed

CPE Observed

CPE Observed

BALB/C Mice, D42

M, 0/14/21

CPE Observed

CPE Observed

CPE Observed

CPE Observed

CPE Observed

CPE Observed

CPE Observed

CPE Observed

CPE Observed

BALB/C Mice, D42

L, 0/14

CPE Observed

CPE Observed

CPE Observed

CPE Observed

CPE Observed

CPE Observed

CPE Observed

CPE Observed

CPE Observed

BALB/C Mice, D42

L, 0/21

CPE Observed

CPE Observed

CPE Observed

CPE Observed

CPE Observed

CPE Observed

CPE Observed

CPE Observed

CPE Observed

BALB/C Mice, D42

L, 0/14/21

CPE Observed

CPE Observed

CPE Observed

CPE Observed

CPE Observed

CPE Observed

CPE Observed

CPE Observed

CPE Observed

BALB/C Mice, D28

H, 0/14

CPE Observed

CPE Observed

CPE Observed

CPE Observed

CPE Observed

CPE Observed

No CPE Observed

CPE Observed

CPE Observed

Guinea pig

, D42

H, 0/21

CPE Observed

CPE Observed

CPE Observed

CPE Observed

CPE Observed

CPE Observed

No CPE Observed

CPE Observed

CPE Observed

Guinea pig

, D42

H, 0/14/21

CPE Observed

CPE Observed

CPE Observed

No CPE Observed

CPE Observed

CPE Observed

No CPE Observed

CPE Observed

CPE Observed

Guinea pig

, D28

L, 0/14

CPE Observed

CPE Observed

CPE Observed

CPE Observed

CPE Observed

CPE Observed

CPE Observed

CPE Observed

CPE Observed

Guinea pig

, D42

L, 0/21

CPE Observed

CPE Observed

CPE Observed

CPE Observed

CPE Observed

CPE Observed

No CPE Observed

CPE Observed

CPE Observed

Guinea pig

, D42

L, 0/14/21

CPE Observed

CPE Observed

CPE Observed

No CPE Observed

CPE Observed

CPE Observed

No CPE Observed

CPE Observed

CPE Observed

Guinea pig

, D28

M, 0/14

CPE Observed

CPE Observed

CPE Observed

CPE Observed

CPE Observed

CPE Observed

CPE Observed

CPE Observed

CPE Observed

Guinea pig

, D42

M, 0/21

CPE Observed

CPE Observed

CPE Observed

CPE Observed

CPE Observed

CPE Observed

CPE Observed

CPE Observed

CPE Observed

Guinea pig

, D42

M, 0/14/21

CPE Observed

CPE Observed

CPE Observed

CPE Observed

CPE Observed

CPE Observed

CPE Observed

CPE Observed

CPE Observed

Rabbit

, D28

H, 0/14

No CPE Observed

CPE Observed

CPE Observed

No CPE Observed

CPE Observed

CPE Observed

No CPE Observed

No CPE Observed

CPE Observed

Rabbit

, D42

H, 0/21

No CPE Observed

CPE Observed

CPE Observed

No CPE Observed

CPE Observed

CPE Observed

No CPE Observed

CPE Observed

CPE Observed

Rabbit

, D42

H, 0/14/21

No CPE Observed

CPE Observed

CPE Observed

No CPE Observed

No CPE Observed

CPE Observed

No CPE Observed

No CPE Observed

CPE Observed

Rabbit

, D28

M, 0/14

No CPE Observed

CPE Observed

CPE Observed

No CPE Observed

CPE Observed

CPE Observed

No CPE Observed

CPE Observed

CPE Observed

Rabbit

, D42

M, 0/21

No CPE Observed

CPE Observed

CPE Observed

No CPE Observed

CPE Observed

CPE Observed

No CPE Observed

CPE Observed

CPE Observed

Rabbit

, D42

M, 0/14/21

No CPE Observed

CPE Observed

CPE Observed

No CPE Observed

CPE Observed

CPE Observed

No CPE Observed

No CPE Observed

CPE Observed

RM

, D28

H, 0/21

No CPE Observed

CPE Observed

CPE Observed

No CPE Observed

No CPE Observed

CPE Observed

No CPE Observed

No CPE Observed

CPE Observed

RM

, D42

H, 0/14

No CPE Observed

CPE Observed

CPE Observed

No CPE Observed

CPE Observed

CPE Observed

No CPE Observed

No CPE Observed

CPE Observed

RM

, D42

H, 0/14

No CPE Observed

CPE Observed

CPE Observed

No CPE Observed

CPE Observed

CPE Observed

No CPE Observed

No CPE Observed

CPE Observed

RM

, D42

H, 0/14

No CPE Observed

CPE Observed

CPE Observed

No CPE Observed

CPE Observed

CPE Observed

No CPE Observed

No CPE Observed

CPE Observed

RM

, D42

H, 0/21

No CPE Observed

CPE Observed

CPE Observed

No CPE Observed

No CPE Observed

CPE Observed

No CPE Observed

No CPE Observed

CPE Observed

RM

, D42

H, 0/21

No CPE Observed

CPE Observed

CPE Observed

No CPE Observed

No CPE Observed

CPE Observed

No CPE Observed

No CPE Observed

CPE Observed

RM

, D42

H, 0/21

No CPE Observed

CPE Observed

CPE Observed

No CPE Observed

No CPE Observed

CPE Observed

No CPE Observed

No CPE Observed

CPE Observed

RM

, D42

H, 0/21

No CPE Observed

CPE Observed

CPE Observed

No CPE Observed

No CPE Observed

CPE Observed

No CPE Observed

No CPE Observed

CPE Observed

RM

, D42

H, 0/14/21

No CPE Observed

No CPE Observed

CPE Observed

No CPE Observed

No CPE Observed

CPE Observed

No CPE Observed

No CPE Observed

No CPE Observed

RM

, D42

H, 0/14/21

No CPE Observed

No CPE Observed

CPE Observed

No CPE Observed

No CPE Observed

CPE Observed

No CPE Observed

No CPE Observed

No CPE Observed

RM

, D42

H, 0/14/21

No CPE Observed

No CPE Observed

CPE Observed

No CPE Observed

No CPE Observed

CPE Observed

No CPE Observed

No CPE Observed

CPE Observed

RM

, D42

H, 0/14/21

No CPE Observed

No CPE Observed

CPE Observed

No CPE Observed

No CPE Observed

CPE Observed

No CPE Observed

No CPE Observed

CPE Observed

In addition, negative and positive controls are not shown for this assay.

The placebo (negative control) is added to the results of Nab assay (Figure-4).

As you know, the Nab assay is like drawing a calibration curve (active virus vs antibody), to find out which concentration of antibody may neutralize the active virus. In case of “cellular immunity”, there is a S-protein peptide that triggers immune cells, which is considered as “positive control”. However, there is not a conventional “positive control”, when humoral antibody of animals is being assayed. Therefore, we have no “Positive control” in Nab assay.

  1. For the ELISA data in Figures 2 and 3, it is not clear what comparisons are being made that are indicated as significant.

Author reply: It is mentioned in figure 2 and 3 caption “The comb-like bars above the columns of the graphs represent the groups of results that came out with significant differences in IgG titration (ELISA).” Now, the following sentences are added to the figure-2 and 3 captions as follows, respectively:

“The star above the comb represents which sera dilutions of comparing categories are different. For example, as what is shown in the top left panel, the IgG (OD) of “Negative control” is considerably less than the interval of “day 0 – 28”, while the IgG (OD) at “day 42” is considerably higher than both of “Negative control” and “day 0 – 28” interval, when the sera was diluted by 1//2112 folds.”

“The star above the comb represents which sera dilutions of comparing categories are different. For example, as what is shown in the top left panel, the IgG (OD) of “Negative control” is considerably less than the interval of “day 0 – 14”, while the IgG (OD) at “day 21-42” interval is considerably higher than both of “Negative control” and “day 0 – 14” interval, when the sera was diluted by 1//2112 folds.”

  1. For the ELISA data, positive controls of sera from infected individuals would be helpful to know how the level of antibody after vaccination compares.

Author reply: That would have been a great index for comparative study. However, the closest method to this idea might have been the “placebo group” in challenge assay, which was only tested in RM cases (and not other animals, since they do not catch a COVID-19 disease). The IgG (OD) in “placebo group” in challenge assay had an amount of almost 0.2 when the sera was diluted either by 132, 528 or 2112 folds, and remained non-significantly the same by the end of challenge assay (6 weeks). Anyway, this fact is now added to the caption of figure-3:

“In case that we consider sera from infected RMs as “positive control”, the IgG (OD) after 132, 528 and 2112 folds of dilution would be 1.9 ± 0.3, 2.1 ± 0.4 and 1.8 ± 0.6, respectively.

  1. The y axis on the ELISA data is labeled as IgG titration (OD), but the unit is just the OD. It should read IgG (OD).

Author reply: That is right. Now, the y axis on the ELISA data is labeled as “IgG (OD)”.

  1. In many of the ELISAs, the serum is not titrated sufficiently.

Author reply:

The following explanation is added to Figure-2 caption:

Further, the following explanation is added to Figure-2 caption:

“The vertical axis “IgG (OD)” represents the optical density (OD) of IgG at 450 nm. The obtained sera were diluted by 132, 528 and 2112 folds, and in cases that the recorded OD was more than one, the samples were diluted again, in a way that their absorbance gets =< 1. Then, the dilution rate (until the absorbance got =< 1) was multiplied by the OD. That is why some of the ODs obtained high extents, such as seven.”

  1. On lines 302-304, they indicate that there were five boosts in antibody levels in the RMs, however, this is not evident in the data. It appears there is only 1 boost in antibody level.

Author reply: Details of how the comb-like bars above the columns of the graphs represent now is mentioned in figure-2 and 3 captions.

Further, last sentences of 5th paragraph of section 3.2 are revised as follows:

“There are two boosts of IgG titration in the RM, as well as Rabbit. However, when high dosage of the candidate vaccine was injected to the RMs via 0/21 mode, five boosts was observed by days 7, 14, 21, 28 and 42 (Figure 3B-left panel).”

  1. Lines 312-313 state that the 0/14/21 regimen scored the highest Nab level in all tested animals, but the graphs for the different groups look identical.

Author reply: That is right. This sentence only applies to RMs. Therefore, the end of aforementioned sentence is rewritten as follows:

“Further, variation of the NAb levels differed by immunization program, in a way that the 0/14/21 regimen scored the highest NAb level in tested RMs.

Reviewer 2 Report

This study examines the efficacy and safety of an inactivated SARS-CoV-2 vaccine in pre-clinical models of infection. The work performed is logical and comprehensive. Data analysis and conclusions are supported by the data.

The work is well performed and, in general, well presented. However, some methods details are lacking, the English is poor in many places, and the test has a large number of minor errors. This detracts from the perceptions of the work and the paper would benefit from correction and further proof reading by the authors.

Major comments

  1. Line 66. The authors state that 5 strains of virus were isolated and pooled. Then further down state that the virus strain that replicated most was chosen for further development. Since no results are presented for the 5 strain pool, what is the point of mentioning the pool?
  2. Line 184/ Lines 221-222.  Was any attempt made to detect live virus in the inactivated vaccine in vitro or in monkeys after vaccination? It is not clear what 'Inactivation of virus by formaldehyde continued up to four batches' means. The reader is directed to Figure 1A. However, Figure 1A provides no data on virus inactivation. Where is the data that shows that the virus was completely inactivated.
  3. Figure 1B. The SDS-PAGE gel shows bands just above 60KDa (annotated N protein) and high molecular weight bands above 160 KDa (annotated S trimer, S dimer). In contrast, the western blot with convalescent serum shows bands above and below 45 KDa and nothing around 60 KDa or above 160 KDa. Please explain what these bands are and how they relate to the N and S structural proteins. Similarly, the band sizes obtained using specific anti-N and anti-S antibodies do not correspond to the 60 KDa and > 160 KDa sizes expected for N and S proteins. There are no obvious bands > 160 KDa with anti-S, and multiple bands ar differing molecular weights with anti-N antibodies. Please explain. Also, the Figure legend states that 'The protein composition of FAKHRAVAC 258 was evaluated by incubating with antibodies targeting N protein'. No mention is made of antibodies targeting the S protein.
  4. Line 278. Figure2. The Y-axes scale from 0 - 7 is labelled as 'IgG titration' (OD)'. Most ELISA readers have a maximum readout between OD 2 and OD 3. What do the numbers 0 - 7 refer to? This should be stated in the Figure legend and Methods section. 

Minor comments

  1. Line 17. Insert 'a' before the word 'global'
  2. Lines 19-21. Replace 'We report the study on potency' with 'We report the results of a study on potency'
  3. Line 27. 'Sever' should be 'severe' Remove the semi-colon between 'patients' and 'have'.
  4. Line 35. Replace 'initiation' with 'the beginning'.
  5. Line 36. Insert 'have been' before 'introduced'.
  6. Line 47. Insert 'In this study' after 'program'.
  7. Line 51. Replace 'here we report the study on potency and safety' with ' Here, we report on the potency and safety'.
  8. Line 52. Insert 'a' after 'in'.
  9. Line 69-70. Insert 'the' before 'rate', and insert 'was' before 'evaluated'.
  10. Line 70-72. Replace 'assigned' with 'equate'.
  11. Line 72-73. Insert 'were' before 'used', and insert 'the' before 'five'.
  12. Figure 1A. Replace 'viruse' with 'virus', and 'formaldehyd' with 'formaldehyde'.
  13. Line 96. Replace '250 milliliters of supernatant was filtered' with '250 milliliters of the supernatant were filtrated'.
  14. Line 97. Insert 'a' between 'to' and 'volume'.
  15. Line 97-98. How long was the virus inactivated with formaldehyde for?
  16. Line 107. Replace 'were' with 'was'.
  17. Insert 'were' before 'employed'.
  18. Line 125. Replace 'loads' with 'preparations' (or other suitable wording).
  19. Line 135. The use of two forward slashes in 1//32, 1//528, etc is unusual and used throughout the manuscript. Normally, one forward slash is used, i.e. 1/32, 1/528. Consider reverting to normal format.
  20. Line 165. What does 'decuple' mean?
  21. Line 188. Replace 'Either of two RMs' with 'Two RMs'.
  22. Line 189. 'via nasal spray of intra-tracheal injection' does not make sense.
  23. Line 206. Replace ' analyzed as a matter of of spike protein' with 'analyzed using spike protein sequences' (or other suitable wording).
  24. Line 185. How were CD4+ and CD8+ T cells enumerated? No methodology is given in Methods section of Figure legends.
  25. Line 220. Replace ' a high replication of virus obtained by 2 x 108 CCID50' with 'high virus replication titers (2 x 108 CCID50) were obtained'.
  26. Line 224-225. What does 'The vaccine product assessed by ELISA technique resulted to more increase in OD450 (specific reactivity with covid-19 patient serum)' mean? What results are being referred to? Also, 'covid-19' should be 'COVID-19' to be consistent with the rest of the manuscript.
  27. Line 235. Replace 'that were show up with a "P" at the beginning' with 'and indicated with a "P" at the beginning'.
  28. Line 243. Replace 'might result to structural' with 'might result in structural'.
  29. Line 245. Replace 'There are three displacements' with 'There are three differences' (or mutations, or substitutions).
  30. Line 265. Two full stops.
  31. Lines 299, 300, 276, 283, 288, 289. '21th ' should be '21st ', and '42th' should be 42nd'.
  32. Line 293. Figure legend. Is statistical significance set a P < 0.05? This should be stated in the Figure legend and the Methods section.
  33. Figure 3 legend. State statistical significance is set at P < 0.05.
  34. Figure 4. Y axes are labelled as 'Nab titration Log2 (1:X)' The scale shown appears to be a titer and not a Log2 transformation. Please comment.
  35. Figure 5 legend does not say how CD4+ and CD8+ T cells were enumerated. This should be set out in the Method section and summarised in the Figure legend. The Y axes have no units. Are these percentages?
  36. Figure 6. Y axes scale not explained in the legend.
  37. Line 365. Two full stops.
  38. Line 365.  '21th' should be '21st'.
  39. Line 369. 'Investigation' should be 'investigation'.
  40. Line 404. 'contraire' should be 'contrary'.

Round 2

Reviewer 1 Report

The authors addressed some of the previous concerns, but several things are still not clear.  One of my major concerns is the OD adjustments for the ELISAs in Figures 2 and 3. The method used to generate these numbers is not clear and potentially misleading. I don’t think it is good to plot adjusted ODs along with actual ODs.  If the sera were titrated down to obtain an OD less than 1, then just plot that titration and OD. So, you will need to plot further dilutions for some of the samples. If you are multiplying the OD by a titration, that value is no longer an OD, it is an adjusted OD. But, if some are adjusted and others are not, they should not be plotted together.

  • On Lines 293-296, the following statement is confusing. The first sentence says the 1/2112 titration remained the same, but the following sentence contradicts that.  When looking at the data, I can’t figure out the point that they are trying to make.

“The IgG titration remained the same by 42 days, when either of 1//132, 1//528 or  1//2112 serum dilutions were tested (Figure 2A, B). However, the IgGs acquired from the sera samples that were diluted 2112 times, came up with considerably different IgG boostlevels (Figure 2 and 3).”

  • The error bars in Figure 2 and 3 are still not clear. According to the description, in Figure 2, according to the description, only the 1/2112 is significant, but it appears that there is a greater difference in the 1/132 and 1/528 dilutions compared to the negative control.  Are all comparisons made to the negative control?

  • What does the double / mean in 1//132? After reading the explanation in the previous review, I still do not understand what the // means.

  • Line 324-327 states that there are five boosts, in RM with the 0/21 mode, but I only observe a boost at d7 and d21. This was also mentioned in the previous review. Maybe they need to clarify what they are defining as a boost.

  • Figure 3 in the revised manuscript is the wrong figure.

  • Line 384-386 states that monitoring the body temperature conferred highly efficient protection in the vaccine groups. This is not evident in Figure 6B. The temperature appears higher in some vaccine groups.

  • In Figure 6C, it appears there may be a difference in weight loss between the placebo and vaccinated males. Is this significant?

  • Line 367-369 states that the viral load after challenge was tested by rtPCR, which confirms higher viral load in the placebo group. These data need to be shown.

Author Response

  1. The authors addressed some of the previous concerns, but several things are still not clear.  One of my major concerns is the OD adjustments for the ELISAs in Figures 2 and 3. The method used to generate these numbers is not clear and potentially misleading. I don’t think it is good to plot adjusted ODs along with actual ODs.  If the sera were titrated down to obtain an OD less than 1, then just plot that titration and OD. So, you will need to plot further dilutions for some of the samples. If you are multiplying the OD by a titration, that value is no longer an OD, it is an adjusted OD. But, if some are adjusted and others are not, they should not be plotted together.

Author reply:

Dear reviewer sorry if the OD results are not explained clearly. Actually, all of the reported ODs in Figure 2 and 3 are adjusted ODs. As you know, according to the Beer–Lambert law, attenuation of light is related to the properties of the material through which the light is travelling. In order to obtain a uniform concentration to the optical path, the concentration and OD must be correlated linearly. The correlation between concentration and OD remains linear as long as the amount of measured OD is less than one. If the measured OD exceeds one, the adjusted OD should be calculated after dilution of the sample and then the OD must be multiplied by a dilution rate. Therefore, the recent adjusted OD would be like actual OD and vice versa.

Besides, we see your concern about “not clear and potentially misleading”. Even if we reported adjusted ODs, it might be expected to have a range of 0-1. To this aim, all the adjusted OD plots are normalized. The method used to generate these numbers is revised in figure-2 and 3 caption, respectively:
